# Theoretical Study on Adsorption Behavior of SF_6_ Decomposition Components on Mg-MOF-74

**DOI:** 10.3390/nano13111705

**Published:** 2023-05-23

**Authors:** Tianxiang Lei, Xiaozhou Fan, Fangcheng Lv, Bowen Jiang

**Affiliations:** 1State Key Laboratory of Alternate Electrical Power System with Renewable Energy Sources, North China Electric Power University, Baoding 071003, China; narroson@126.com (T.L.); lfc@ncepu.edu.cn (F.L.); qq953501277@icloud.com (B.J.); 2Hebei Provincial Key Laboratory of Power Transmission Equipment Security Defence, North China Electric Power University, Baoding 071003, China

**Keywords:** SF_6_ decomposition species, gas sensing material, MOFs, principle of quantum chemistry

## Abstract

SF_6_ gas is an arc extinguishing medium that is widely used in gas insulated switchgear (GIS). When insulation failure occurs in GIS, it leads to the decomposition of SF_6_ in partial discharge (PD) and other environments. The detection of the main decomposition components of SF_6_ is an effective method to diagnose the type and degree of discharge fault. In this paper, Mg-MOF-74 is proposed as a gas sensing nanomaterial for detecting the main decomposition components of SF_6_. The adsorption of SF_6_, CF_4_, CS_2_, H_2_S, SO_2_, SO_2_F_2_ and SOF_2_ on Mg-MOF-74 was calculated by Gaussian16 simulation software based on density functional theory. The analysis includes parameters of the adsorption process such as binding energy, charge transfer, and adsorption distance, as well as the change in bond length, bond angle, density of states, and frontier orbital of the gas molecules. The results show that Mg-MOF-74 has different degrees of adsorption for seven gases, and chemical adsorption will lead to changes in the conductivity of the system; therefore, it can be used as a gas sensing material for the preparation of SF6 decomposition component gas sensors.

## 1. Introduction

Sulfur hexafluoride gas is an arc extinguishing medium widely used in gas insulated switch gear (GIS). However, insulation defects inside GIS can lead to partial discharge (PD) and other faults. SF_6_ decomposes into extremely unstable low-fluorine sulfides (SF_n_, n = 1~5) in an overheated environment with long-term failure [1,2,3,4,5]. Although SF_n_ can collide with F atoms in the environment to recover to SF_6_ molecules, in the presence of micro-amounts of O_2_ and H_2_O, SF_n_ further reacts to form SO_2_, SOF_2_, SO_2_F_2_, H_2_S and other major products [6,7,8]; if the fault occurs on the basin insulator, SF_n_ can react with electrode materials, insulation materials, etc., to generate characteristic gases CF_4_, CS_2_, etc. [9,10]; therefore, the types of the above decomposition products are closely related to the fault types. Detecting the main decomposition components of SF_6_ in GIS equipment by the gas sensor method is an effective method to diagnose discharge faults and types [11,12,13,14], and guarantees the safe operation of electrical insulation equipment.

The core of gas sensors is gas sensitive material. At present, the gas sensitive materials used to detect SF_6_ decomposition gas mainly include: noble metal surface modified inorganic materials (including metal oxides [15,16,17,18,19,20], metal sulfides [21,22,23,24], metal selenides [25,26,27,28]) series; metal-doped carbon-based materials (including graphene [29,30,31,32,33], carbon nanotubes [34,35,36,37]) series; and composite materials [38,39,40] compounded by inorganic materials and carbon-based materials. These modified nanomaterials exhibit good gas sensing properties, but the limited design and flexibility of inorganic materials limit the development space of the above materials.

Metal-organic frameworks (MOFs) are inorganic-organic hybrid crystalline porous materials formed by coordination bonds among metal ions or metal clusters as nodes and organic ligands [41]; they have the characteristics of design flexibility, structural diversity, adjustable pore size and high specific surface area. Therefore, by changing the types of metal ions or metal clusters and the type, structure and chain length of organic ligands, the pore size and spatial morphology of MOFs can be controlled to synthesize MOFs materials with different three-dimensional structures and different pore sizes [42,43,44,45]. These unique and novel materials have been rapidly developed in the past two decades, and have become an ideal platform for various advanced functional materials and applications, making them important in the aspects of gas storage, adsorption and separation, catalysis, analysis and detection, drug delivery and so on [46,47,48,49,50]. However, the research and application of gas sensitive materials for detecting the main decomposition components of SF_6_ are rarely reported.

In the study of inorganic metal oxide and sulfide gas sensing materials, Peng, S. et al. first studied the gas sensing properties of metal oxide (ZnO) gas sensors for SF_6_ decomposition components (SO_2_, SOF_2_, SO_2_F_2_). The results showed that the response of flower-like ZnO to SO_2_ is better at 270–300 °C, while the response of SOF_2_ and SO_2_F_2_ is weaker [51]. Li, B.L. and other scholars have studied scandium-doped molybdenum sulfide materials (Sc-MoS_2_). Based on density functional theory, the adsorption properties of intrinsic MoS_2_ and Sc-MoS_2_ materials on four typical SF_6_ decomposition gas molecules (H_2_S, SO_2_, SOF_2_, SO_2_F_2_) were studied [52]. In the study of gas adsorption in metal-organic frameworks, Lee, K. et al. studied the Van der Waals adsorption of isomorphic M-MOF-74 (M = Mg, Ti, V, Cr, Mn, Fe, Co, Ni, Cu, Zn) on H_2_, CO, CO_2_, H_2_O, H_2_S, N_2_, NH_3_, and SO_2_ based on the DFT principle [53]. Liu, J. et al. studied the adsorption properties of 11 MOFs materials (including Mg-MOF-74) for H_2_S, and then separated H_2_S from H_2_S/CO_2_ [54]. Li, S. et al. studied and compared the adsorption parameters of benzene ring and Cu on H_2_S and SO_2_ gas molecules in MOF-505, and the results showed that the adsorption of gas molecules on Cu was greater than that on benzene ring [55].

Mg-MOF-74 (CPO-27-Mg) is a new type of nanoscale MOFs material with Mg^2+^ as the central ion and 2,5-dihydroxy-1,4-benzenedicarboxylic acid as the ligand. Henkelis, S.E. et al. introduced the synthesis method of this MOF in detail [56]. The crystal structure used in this article is generated by single crystal X-ray analysis, and is currently recorded in the Cambridge structure database (doi: 10.5517/ccdc.csd.cc20k4pf). By analyzing the crystal information file (.cif) in CCDC 1863524, the metal ions/ligands ratio in a single unit cell of Mg-MOF-74 is 9:7. Another useful method for calculating the metal ions/ligands ratio which was mentioned by Visa, A. et al. in the article [57]: the relationship of the ratio between the number of metal ions and ligands can be deducted by gradually building a 3D supramolecular network throughout the HyperChem 7.52 package.

As a gas sensing material, the pore size of 1.2 nm (4 × 4 × 4 super cell shown in Figure 1) provides the structural premise for SF_6_, CF_4_, CS_2_, H_2_S, SO_2_, SO_2_F_2_ and SOF_2_ to enter the pores of the material. The unsaturated coordinated Mg^2+^ in the pores provides the possibility for gas adsorption. In this paper, through the simulation research based on the DFT principle, by selecting a more accurate functional, the chemical adsorption characteristics of Mg-MOF-74 clusters on SF_6_ and CF_4_, CS_2_, H_2_S, SO_2_, SO_2_F_2_ and SOF_2_ produced by SF_6_ discharge decomposition in GIS equipment are analyzed more accurately. By introducing the frontier molecular orbital theory, the energy gap change of the cluster after adsorbing gas is analyzed, and then the influence of adsorption behavior on the conductivity of the material is speculated. The application prospect of Mg-MOF-74 as a gas sensing material for on-line detection of SF_6_ discharge decomposition products is predicted theoretically. It provides a new perspective for the research and application of MOFs as new gas sensitive materials.

## 2. Calculation Parameter Setting and Model Construction

The modeling involved in this paper is completed in GaussView, and the structural optimization and single point calculation are completed in Gaussian 16 software. In the application of Gaussian series quantum chemistry simulation software, Sciortino, G. et al. found that the PBE0 functional and the def2-tzvp basis set have extremely high accuracy in the calculation of Ni(II) complexes [58]. PBE0 functional together with def-tzvp is the best-performing method, and is excellent in the study of platinum-catalyzed chemical reaction mechanism based on DFT principle [59].

Therefore, when dealing with the exchange correlation term of electrons, we also use the PBE0 hybrid functional with higher calculation accuracy than LDA, GGA and meta-GGA. Because the sensitivity of geometric structure optimization to the basis set is much lower than that of single point energy calculation, and the time consumption of these tasks is much higher than that of single point calculation, the def2-svp (2-zeta) basis set with appropriate accuracy is selected for structural optimization, and the def2-tzvp (3-zeta) basis set with higher accuracy is used in single point energy calculation; the GD3BJ algorithm is used to correct the Van der Waals effect, and sets the charge to 0 and the spin multiplicity to 1.

By reading the crystallographic information (.cif) file, each periodic structural unit of Mg-MOF-74 contains 638 atoms, 704 chemical bonds and 18 polyhedra. Due to the application of high-precision functional, the computational power cannot meet the optimization of the overall periodic structure and the single point calculation of adsorption. It can be seen from Figure 2 that the smallest repeating unit in Mg-MOF-74 is a cluster containing four Mg atoms, three benzene rings, three hydroxyl groups, three carboxyl groups and one coordinated water (inside the white dotted box). Therefore, this paper uses this cluster (without bound water) as the adsorption substrate to simulate the adsorption characteristics of seven gas molecules on this segment, and then infers the macroscopic adsorption characteristics of the material on gases.

The optimized Mg-MOF-74 cluster and gas molecular model are shown in Figure 3. In order to represent the total energy change of gas molecules adsorbed on Mg-MOF-74, the binding energy during adsorption is defined as:*E_binding_* = *E_MOF-gas_* − *E_MOF_* − *E_gas_*- + *E_BSSE_*,(1)

In Equation (1): *E_MOF-gas_* is the total energy of the system after Mg-MOF-74 adsorbs gas molecules, *E_MOF_* is the total energy of Mg-MOF-74 clusters before adsorption, and *E_gas_* is the total energy of gas molecules before adsorption. The basis set superposition error (BSSE) is corrected using the counterpoise method proposed by Boys and Bernardi [60], and *E_BSSE_* is the correction value.

The charge transfer amount is the number of charge transfer obtained by analyzing the charge population through the Hirshfeld charge model.
Δ*Q* = *Q*_1_ − *Q*_2_,(2)

In Equation (2): Δ*Q* is the charge transfer amount of the system, *Q*_1_ is the charge of gas molecules after adsorption, and *Q*_2_ is the charge of gas molecules before adsorption.

The adsorption distance is defined as the distance between the gas molecule and the adsorption site of Mg-MOF-74. The Van der Waals radius is 1/2 of the distance between two adjacent nuclei when atoms interact with each other through Van der Waals force. The covalent radius is 1/2 of the nucleus spacing when the atoms of the same element form diatomic molecules.

The change of charge density is analyzed by the distribution of the yellow region (atom has electron loss property) and the blue region (atom has electron gain property) in the differential charge density diagram.

In this paper, the discrete orbital occupation diagram is broadened by Gaussian function to obtain the density of states (DOS) curve, and the chemical adsorption of Mg-MOF-74 on gas is further analyzed by the total density of states, gas density of states and local density of states.

## 3. Results

The adsorption calculation in this paper makes the SF_6_, CF_4_, CS_2_, H_2_S, SO_2_, SO_2_F_2_ andSOF_2_ gas molecules vertically close to the unsaturated sites on the Mg-MOF-74 material; after reaching the most stable state, by extracting the adsorption parameters (binding energy, charge transfer amount, adsorption distance), the change of bond length and bond angle of gas analysis after adsorption is measured, and the adsorption capacity of Mg-MOF-74 to seven gases is comprehensively judged by analyzing the change of orbital occupancy.

### 3.1. Parameters of Adsorption Behavior

Adsorption Model:

The adsorption model of SF_6_, CF_4_, CS_2_, H_2_S, SO_2_, SO_2_F_2_ and SOF_2_ gas molecules on Mg-MOF-74 material is shown in Figure 3a–g.

It can be seen that the F atoms in CF_4_ and SF_6_ interact with the Mg atoms in the adsorbed substrate, Figure 4a,b. The S atoms in CS_2_ and H_2_S interact with the Mg atoms in the adsorption substrate, Figure 3c,d. The O atom and S atom in SO_2_ gas interact with the Mg atom and O atom in the adsorbed substrate, respectively, Figure 3e. The O atoms in SO_2_F_2_ and SOF_2_ interact with the Mg atoms in the adsorption substrate, Figure 3f,g.

2.Parameters of adsorption behavior:

The adsorption energies, charge transfer and adsorption distance in the adsorption process of seven gas molecules SF_6_, CF_4_, CS_2_, H_2_S, SO_2_, SO_2_F_2_ and SOF_2_ on Mg-MOF-74 are listed in Table 1.

Combined with Formula (1), the adsorption energy of Mg-MOF-74 material to each gas molecule is less than 0, indicating that the adsorption process system reaches a lower energy stable state after heat-releasing; the relationship among the adsorption capacity of the material to each gas is: H_2_S > SO_2_ > SOF_2_ > SO_2_F_2_ > CS_2_ > SF_6_ > CF_4_.

According to Formula (2) in this paper, the gas molecules lose electrons during the adsorption process, and the substrate material Mg-MOF-74 obtains electrons.

The adsorption distance among each gas molecule and the substrate is less than the sum of Van der Waals radius and larger than the sum of covalent radius (shown in Appendix A). Therefore, according to the adsorption distance, it can be inferred that the strength of the chemical bond formed by the adsorption of seven gases by Mg atoms on Mg-MOF-74 material belongs to weaker chemical action.

3.Differential charge density:

Figure 5a–g shows the charge density difference in the adsorption process. The yellow regions show the electron losing property and the blue regions show the electron gaining property.

The charge distribution near the F atom of SF_6_ and CF_4_ is relatively uniform, and the charge distribution inside the molecule does not change significantly. The charge distribution near the C atom in CS_2_ is uniform, and the Mg atom bonded to the S atom in CS_2_ is wrapped by the yellow regions. The charge distribution near the H atom in H_2_S is uniform, and the Mg atom bonded to the S atom in H_2_S is surrounded by the yellow regions. The Mg atom bonded to the O atom in SO_2_ is wrapped by the yellow regions. The charge distribution near the S atom and F atom in SO_2_F_2_ is uniform, and the Mg atom bonded to the O atom in SO_2_F_2_ is wrapped by the yellow regions. The charge distribution near the S and F atoms in SOF_2_ is uniform, and the Mg atom bonded to the O atom in SOF_2_ is wrapped by the yellow regions. The above results show that the Mg atoms on the Mg-MOF-74 material exhibit electron-gaining properties during the adsorption process, while the gas molecules exhibit electron-losing properties.

### 3.2. The Change of Bond Length and Bond Angle of Gas Molecules after Adsorption

The bond length changes of SF_6_, CF_4_, CS_2_, H_2_S, SO_2_, SO_2_F_2_ and SOF_2_ molecules before and after adsorption and the bond angle changes are shown in Appendix A.

The bond length of SF_6_, CF_4_, CS_2_, H_2_S, SO_2_, SO_2_F_2_ and SOF_2_ gas molecules changes slightly due to adsorption; the obvious changes in the bond angle (atomic number shown in Figure 2b–g) are as follow: SF_6_ molecules F(1)-S(1)-F(4) and F(2)-S(1)-F(3) decrease by 1.28° and 1.20°, respectively; CF_4_ molecules F(1)-C(1)-F(3) decrease by 1.28°; SO_2_ molecules O(1)-S(1)-O(2) decrease by 2.13°; SO_2_F_2_ molecules F(1)-S(1)-O(2) decrease by 2.2°; SOF_2_ molecules F(1)-S(1)-F(2) increase by 1°; for CS_2_ and H_2_S, the bond angle change is less than 1°. All the bond length and bong angle changes are shown in Appendix A.

The changes of bond length and bond angle before and after adsorption of the above seven gas molecules, together with the adsorption energy, adsorption distance, charge transfer amount and differential charge density diagram, strongly prove that there are certain interactions among gas molecules and the adsorption substrate.

### 3.3. The Orbital Occupation Changes of Each System before and after Gas Adsorption

The orbital occupation calculation results of Gaussian 16 software are imported into Multiwfn software to obtain discrete orbital occupation information under different energies, as indicated by the blue arrow in Figure 6a.

The Gaussian function is used for broadening, and the half-peak width is set to 0.01 eV to obtain a continuous density of states diagram (as indicated by the green arrow in Figure 6b, which can clearly and intuitively analyze the orbital occupation change. According to the extra-nuclear electron arrangement of Mg atoms, there are no filled electrons in the 3p orbital of Mg atoms, and Mg atoms form Mg^2+^ by losing two electrons in the 3s orbital. Therefore, the bonding interaction during adsorption is analyzed by the overlap of the 3s and 3p orbital curves of Mg atoms on Mg-MOF-74 material with the outer orbital curves of SF_6_, CF_4_, CS_2_, H_2_S, SO_2_, SO_2_F_2_ and SOF_2_ gas molecules.

The orbital occupation of Mg-MOF-74 after adsorbing SF_6_ gas is shown in Figure 7a–c.

By observing Figure 7a, it can be found that after adsorbing SF_6_ gas, a new orbital occupation appears near the position of −1.95 eV, and the other energy positions do not change significantly. Compared with the total orbital occupation, it can be seen that the orbital occupation changes near the energy of −1.95 eV, 0.44 eV and 1.65 eV after adsorption of SF_6_ are contributed by SF_6_, as shown in Figure 7b. By analyzing the 3s, 3p orbitals of the adsorbed substrate Mg atom and the 2p orbital occupation of the SF_6_ gas molecule F atom, it can be clearly found that the 2p orbital of the F atom and the 3s orbital occupation broadening curve of the Mg atom overlap near the energy −1.95 eV position (as shown in Figure 7c), which indicates that the Mg-MOF-74 material has bonding adsorption effect on the SF_6_ gas.

2.The orbital occupation of Mg-MOF-74 after adsorbing CF_4_ gas is shown in Figure 8a–c.

By observing Figure 8a, it can be found that there is no new orbital occupation after adsorption of CF_4_ gas. Compared with the total orbital occupation, it can be seen that the orbital occupation changes near the energy of 0.42 eV, 1.16 eV and 2.21 eV after CF_4_ adsorption are contributed by CF_4_, as shown in Figure 8b.

By analyzing the 3s and 3p orbitals of the Mg atom of the adsorption substrate and the 2p orbital occupation of the F atom of the CF_4_ gas molecule, it can be clearly found that the 2p orbital of the F atom does not overlap with the 3s and 3p orbital occupation broadening curves of the Mg atom (as shown in Figure 8c), which indicates that the Mg-MOF-74 material barely has bonding adsorption effect on the CF_4_ gas.

3.The orbital occupation of Mg-MOF-74 after adsorbing CS_2_ gas is shown in Figure 9a–c.

It can be seen from Figure 9a that after adsorbing CS_2_ gas, a new orbital occupation appears near the position of −2.03 eV, and the other energy positions do not change significantly. Compared with the total orbital occupancy, it can be seen that the orbital occupancy changes near the energy of −2.03 eV, 0.84 eV and 3.69 eV after adsorption of CS_2_ are contributed by CS_2_, as shown in Figure 9b. By analyzing the 3s and 3p orbitals of the Mg atom of the adsorption substrate and the 3p orbital occupation of the S atom of the CS_2_ gas molecule, it can be clearly found that the 3p orbital of the S atom and the 3p orbital occupation broadening curve of the Mg atom overlap near the energy −2.03 eV position (as shown in Figure 9c), which indicates that the Mg-MOF-74 material has bonding adsorption effect on the CS_2_ gas.

4.The orbital occupation of Mg-MOF-74 after adsorbing H_2_S gas is shown in Figure 10a–c.

It can be found from Figure 10a that there is no new orbital occupation after adsorption of H_2_S gas. Compared with the total orbital occupation, it can be seen that the orbital occupation changes of the system near the energy of 0.35 eV, −6.95 eV and −7.47 eV after adsorption of H_2_S are contributed by H_2_S, as shown in Figure 10b.

By analyzing the 3s and 3p orbitals of the Mg atom of the adsorption substrate and the 3p orbital occupation of the S atom of the H_2_S gas molecule, it can be clearly found that the 3p orbital of the S atom and the 3s orbital occupation broadening curve of the Mg atom overlap near the energy of 0.35 eV (as shown in Figure 10c), which indicates that the Mg-MOF-74 material has bonding adsorption effect on the H_2_S gas.

5.The orbital occupation of Mg-MOF-74 after adsorbing SO_2_ gas is shown in Figure 11a–c.

It can be seen from Figure 11a that after the adsorption of SO_2_ gas, a new orbital occupation occurs near −3.56 eV, and the remaining energy positions do not change significantly. Compared with the total orbital occupation, it can be seen that the change of orbital occupation near the energy of −3.56 eV, −10.35 eV and 1.38 eV after SO_2_ adsorption is contributed by SO_2_, as shown in Figure 11b. By analyzing the 3s and 3p orbitals of the Mg atom of the adsorption substrate and the 2p orbital occupation of the O atom of the SO_2_ gas molecule, it can be clearly found that the 2p orbital of the O atom and the 3p orbital occupation broadening curve of the Mg atom overlap near the energy −3.56 eV position (as shown in Figure 11c), which indicates that the Mg-MOF-74 material has bonding adsorption effect on the SO_2_ gas.

6.The orbital occupation of Mg-MOF-74 after adsorbing SO_2_F_2_ gas is shown in Figure 12a–c.

By observing Figure 11a, it can be found that there is no new orbital occupation after adsorption of SO_2_F_2_ gas. Compared with the total orbital occupation, it can be seen that the change of orbital occupation near the energy of −1.02 eV, −0.1 eV and 0.89 eV after adsorption of SO_2_F_2_ is contributed by SO_2_F_2_, as shown in Figure 12b.

By analyzing the 3s and 3p orbitals of the Mg atom of the adsorption substrate and the 2p orbital occupation of the O atom of the SO_2_F_2_ gas molecule, it can be clearly found that the 2p orbital of the O atom and the 3p orbital occupation broadening curve of the Mg atom overlap near the energy −1.02 eV position (as shown in Figure 12c), which indicates that the Mg-MOF-74 material has bonding adsorption effect on the SO_2_F_2_ gas.

7.The orbital occupation of Mg-MOF-74 after adsorbing SOF_2_ gas is shown in Figure 13a–c.

It can be found from Figure 13a that after the adsorption of SOF_2_ gas, a new orbital occupation appears near −2.07 eV after adsorption, and the other energy positions do not change significantly. Compared with the total orbital occupation, it can be seen that the orbital occupation changes near the energy of −2.07 eV, −0.93 eV and 0.77 eV after adsorption of SOF_2_ are contributed by SOF_2_, as shown in Figure 13b.

By analyzing the 3s and 3p orbitals of the Mg atom of the adsorption substrate and the 2p orbital occupation of the O atom of the SOF_2_ gas molecule, it can be clearly found that the 2p orbital of the O atom and the 3p orbital occupation broadening curve of the Mg atom overlap near the energy −0.93 eV position (as shown in Figure 13c), which indicates that the Mg-MOF-74 material has bonding adsorption effect on the SOF_2_ gas.

### 3.4. Conductivity Analysis after Adsorption

Pham, H.Q et al. have systematically studied the electronic band structure of a series of reticular metal-organic framework materials based on density functional theory [61]. By calculating the HOMO-LUMO gap under different types, different numbers of substituents and different C_Ar_-C_Ar_-C=O dihedral angle models, it is revealed that the band gap energy can be predicted by the HOMO-LUMO gap of MOFs organic ligands. The results show that the electronic band structure of MOFs can be calculated by first-principles calculations of organic linkers instead of complex and time-consuming calculations on periodic systems. In this section, by introducing the frontier molecular orbital theory to calculate the energy gap, the conductivity change of the cluster after adsorbing gas is qualitatively analyzed. It indicates the effect of gas adsorption on the conductivity of Mg-MOF-74 material.

The frontier molecular orbital distribution and energy of Mg-MOF-74 before and after adsorption of seven gases are shown in Figure 14. From the diagram, it can be seen that the highest occupied molecular orbital (HOMO) of the Mg-MOF-74 cluster and the system after adsorbing SF_6_, CF_4_, CS_2_, H_2_S, SO_2_, SO_2_F_2_ and SOF_2_ are mainly distributed on the surface of the benzene ring, which is also the same as the law in the literature [61]. The lowest unoccupied molecular orbital (LUMO) of SF_6_, CS_2_, SO_2_ and SO_2_F_2_ systems are mainly distributed on the surface of gas molecules. The LUMO of CF_4_, H_2_S and SO_2_F_2_ systems are mainly distributed on the surface of the other benzene ring of the cluster. With the change of frontier orbital distribution, the HOMO and LUMO energies of the seven systems have different degrees of increase and decrease, respectively. For HOMO, the change of SO_2_F_2_ system is the most obvious, and for LUMO, the change of SO_2_ system is the most obvious.

The energy gap’s change demonstrates that when the Mg-MOF-74 cluster only has chemical adsorption for CF_4_, SF_6_, CS_2_, H_2_S, SO_2_, SO_2_F_2_ and SOF_2_ gases, the conductivity of each system has different degrees of improvement, and the order of promotion is: SO_2_ > CS_2_ > SOF_2_ > SF_6_ > SO_2_F_2_ > CF_4_ > H_2_S. Therefore, in the application, it is possible to analyze the gas composition by comparing the resistance value response difference of the material to seven gases under the same sensor electrode preparation, the same gas flow rate, and the same measurement temperature, and then determine whether SF_6_ decomposes or not.

## 4. Conclusions

In this paper, GaussView software is used to construct the cluster of Mg-MOF-74 material and seven gas molecular models of SF_6_, CF_4_, CS_2_, H_2_S, SO_2_, SO_2_F_2_ and SOF_2_. The adsorption properties of Mg-MOF-74 clusters to gases are calculated by Gaussian16 software based on the DFT principle. The main conclusions are as follows:

When the adsorption of each gas molecule by Mg-MOF-74 material reaches a stable state, the relationship among the adsorption capacity of each gas is: H_2_S > SO_2_ > SOF_2_ > SO_2_F_2_ > CS_2_ > SF_6_ > CF_4_.During the adsorption process, the metal atoms in the MOFs gain electrons, the gas molecules lose electrons, and the adsorption process leads to changes in the bond length and bond angle of the gas molecules.

The density of states curve obtained by orbital occupation broadening shows that the adsorption of CF_4_ by Mg-MOF-74 belongs to physical adsorption, and the adsorption of the other six gases belongs to chemical adsorption.

The frontier molecular orbital analysis shows that the chemical adsorption of Mg-MOF-74 on gas causes the change of conductivity of the system. Therefore, the response difference of the material to the gas in GIS and pure SF_6_ gas can be used to determine whether there is a fault inside the equipment.

## Figures and Tables

**Figure 1 nanomaterials-13-01705-f001:**
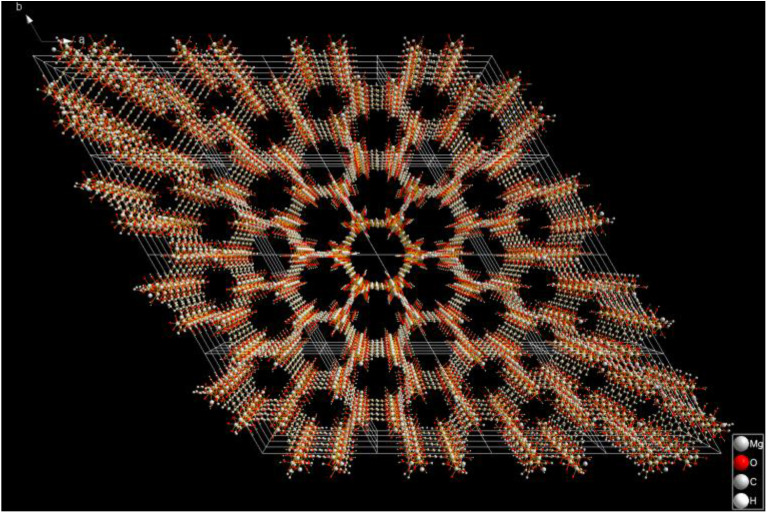
4 × 4 × 4 super cell of Mg-MOF-74.

**Figure 2 nanomaterials-13-01705-f002:**
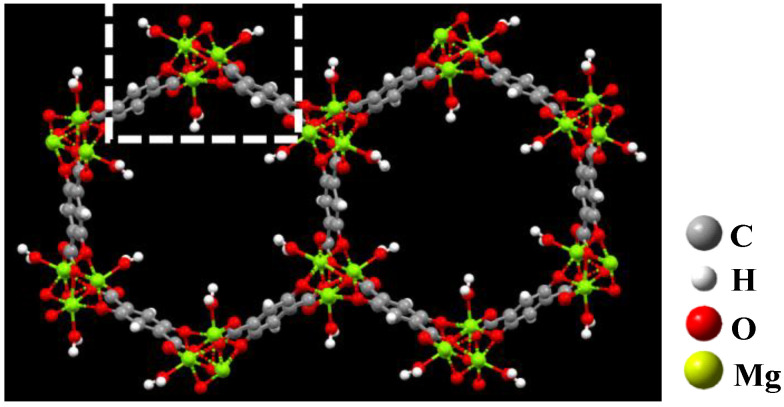
The extracted Mg-MOF-74 clusters.

**Figure 3 nanomaterials-13-01705-f003:**
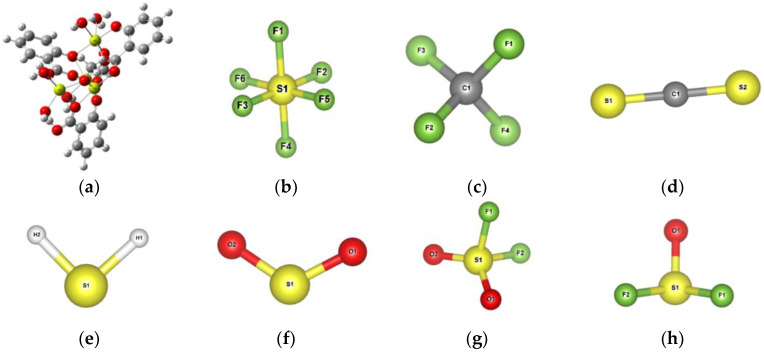
Mg-MOF-74 clusters and gas molecular models after structure optimization. (**a**) Mg-MOF-74clusters, (**b**) SF_6_, (**c**) CF_4_, (**d**) CS_2_, (**e**) H_2_S, (**f**) SO_2_, (**g**) SO_2_F_2_, (**h**) SOF_2_.

**Figure 4 nanomaterials-13-01705-f004:**
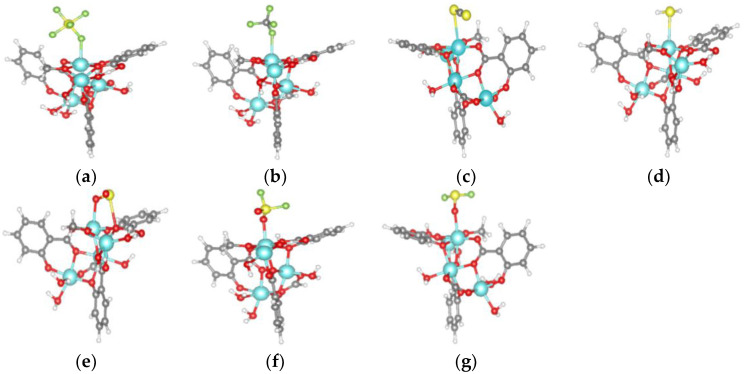
Adsorption models of gas molecules on Mg-MOF-74 material. (**a**) SF_6_, (**b**) CF_4_, (**c**) CS_2_, (**d**) H_2_S, (**e**) SO_2_, (**f**) SO_2_F_2_, (**g**) SOF_2_.

**Figure 5 nanomaterials-13-01705-f005:**
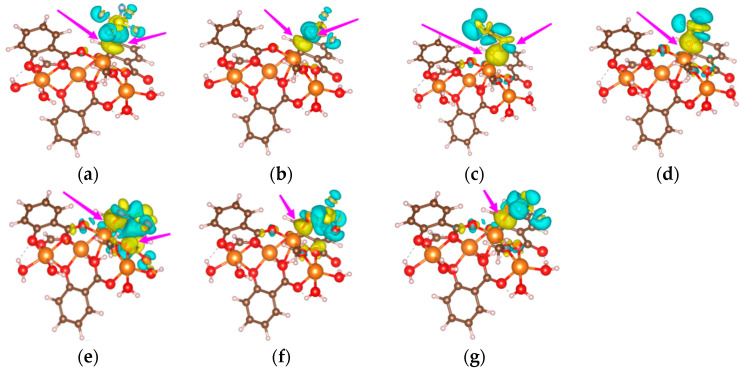
Differential charge density diagrams of gas molecules adsorbed on Mg-MOF-74 material. (**a**) SF_6_, (**b**) CF_4_, (**c**) CS_2_, (**d**) H_2_S, (**e**) SO_2_, (**f**) SO_2_F_2_, (**g**) SOF_2_.

**Figure 6 nanomaterials-13-01705-f006:**
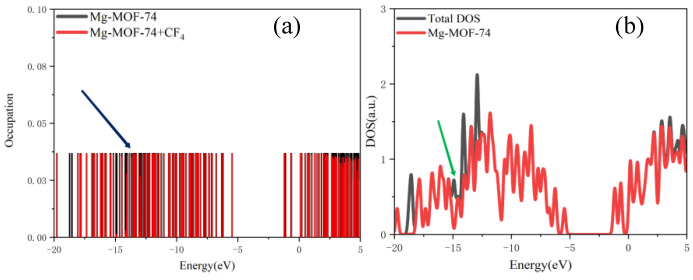
(**a**) Discrete orbital occupation information, (**b**) broadening.

**Figure 7 nanomaterials-13-01705-f007:**
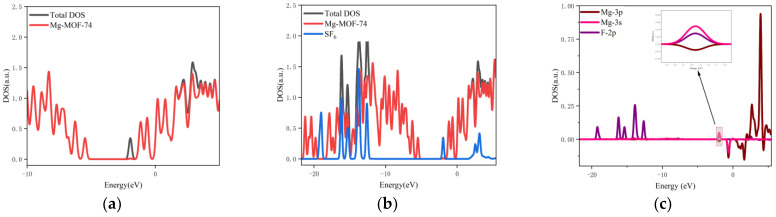
The orbital occupancy information of Mg-MOF-74 after adsorbing SF_6._ (**a**) The orbital occupation comparison of Mg-MOF-74 after adsorbing SF_6_, (**b**) Comparison of orbital occupancy between materials and gas, (**c**) The occupation analysis of 3s,3p orbital of Magnesium atom and 2p orbital of Fluorine atom.

**Figure 8 nanomaterials-13-01705-f008:**
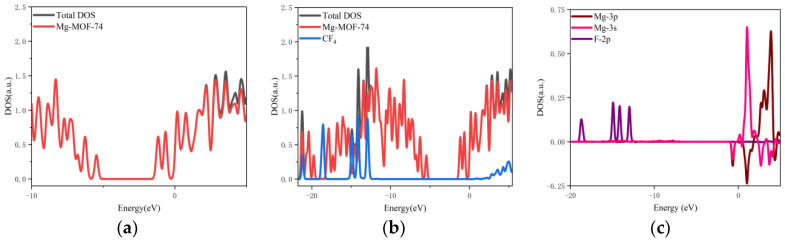
The orbital occupancy information of Mg-MOF-74 after adsorbing CF_4_. (**a**) The orbital occupation comparison of Mg-MOF-74 after adsorbing CF_4_, (**b**) Comparison of orbital occupancy between materials and gas, (**c**)The occupation analysis of 3s,3p orbital of Magnesium atom and 2p orbital of Fluorine atom.

**Figure 9 nanomaterials-13-01705-f009:**
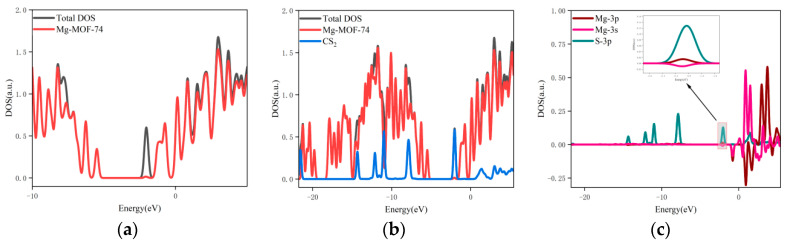
The orbital occupancy information of Mg-MOF-74 after adsorbing CS_2_. (**a**) The orbital occupation comparison of Mg-MOF-74 after adsorbing CS_2_, (**b**) Comparison of orbital occupancy between materials and gas, (**c**) The occupation analysis of 3s,3p orbital of Magnesium atom and 3p orbital of Sulfur atom.

**Figure 10 nanomaterials-13-01705-f010:**
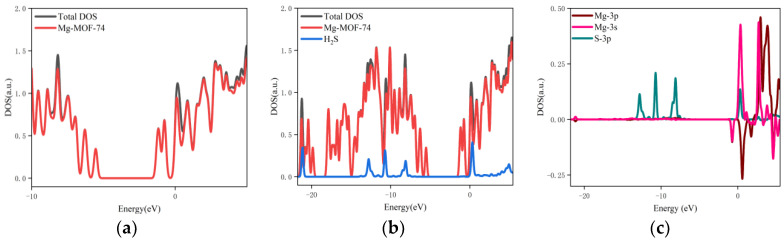
The orbital occupancy information of Mg-MOF-74 after adsorbing H_2_S. (**a**) The orbital occupation comparison of Mg-MOF-74 after adsorbing H_2_S, (**b**) Comparison of orbital occupancy between materials and gas, (**c**) The occupation analysis of 3s,3p orbital of Magnesium atom and 3p orbital of Sulfur atom.

**Figure 11 nanomaterials-13-01705-f011:**
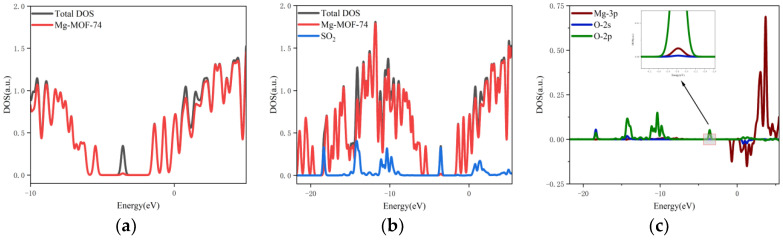
The orbital occupancy information of Mg-MOF-74 after adsorbing SO_2_. (**a**) The orbital occupation comparison of Mg-MOF-74 after adsorbing SO_2_, (**b**) Comparison of orbital occupancy between materials and gas, (**c**)The occupation analysis of 3p orbital of Magnesium atom and 2s, 2p orbital of Oxygen atom.

**Figure 12 nanomaterials-13-01705-f012:**
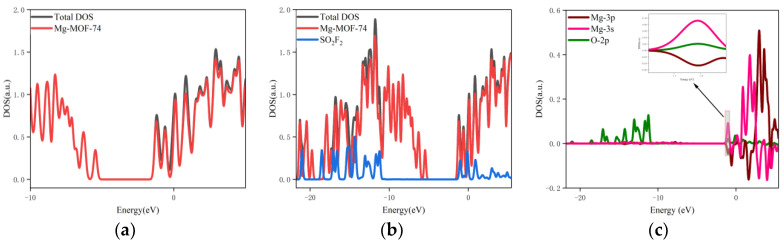
The orbital occupancy information of Mg-MOF-74 after adsorbing SO_2_F_2_. (**a**) The orbital occupation comparison of Mg-MOF-74 after adsorbing SO_2_F_2_, (**b**) Comparison of orbital occupancy between materials and gas, (**c**)The occupation analysis of 3s,3p orbital of Magnesium atom and 2p orbital of Oxygen atom.

**Figure 13 nanomaterials-13-01705-f013:**
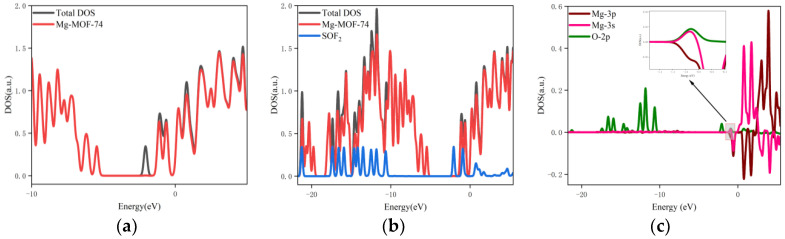
The orbital occupancy information of Mg-MOF-74 after adsorbing SOF_2._ (**a**) The orbital occupation comparison of Mg-MOF-74 after adsorbing SOF_2_, (**b**) Comparison of orbital occupancy between materials and gas, (**c**)The occupation analysis of 3s,3p orbital of Magnesium atom and 2p orbital of Oxygen atom.

**Figure 14 nanomaterials-13-01705-f014:**
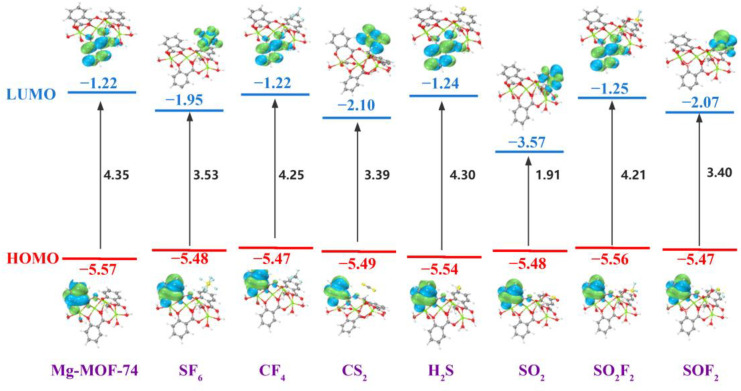
The frontier molecular orbital distribution of Mg-MOF-74 after adsorption gas.

**Table 1 nanomaterials-13-01705-t001:** The adsorption energy and charge transfer amount of Mg-MOF-74 for gas molecules.

Gas Mode	SF_6_	CF_4_	CS_2_	H_2_S	SO_2_	SO_2_F_2_	SOF_2_
Adsorption energy (eV)	−0.208	−0.179	−0.296	−0.474	−0.425	−0.363	−0.418
Charge transfer amount (e)	0.127	0.145	0.148	0.173	0.086	0.174	0.217
Adsorption distance (Å)	2.342	2.374	2.961	2.783	2.252	2.281	2.212

## Data Availability

The data presented in this study is available on request from the corresponding authors.

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
