# Peer review of "Theoretical Study on Adsorption Behavior of SF6 Decomposition Components on Mg-MOF-74"

_nanomaterials, 2023, doi:10.3390/nano13111705_

Round 1
Reviewer 1 Report
The article entitled "Research on the adsorption behaviour of SF6 decomposition components by Mg-MOF-74" describes a very interesting application by Mg-MOF-74. The paper focuses on the absorption of different gases as appreciated by quantochemical calculations using Gaussian 16 simulation software. Although extensive work has been performed, several points must be improved and the manuscript rearranged before acceptance.
1. Please clearly mention in the last paragraph of the introduction what the novelty of your work is, including examples of different MOFs suitable for this application.
2. It is necessary to give more information about the investigated Mg-MOF-74, such as surface volume, pore size, and the ligand used since the inorganic node is Mg.
3. The number of tables is huge, and in the 3.2 subparagraph, I got lost in so much information. Please compress Table 4 and Table 5, since they refer to the same gas in only one table with the values before and after adsorption. And so on in the entire subsection for the seven gases that you analyze.
4. It is necessary to use the same number of digits for the table values. I recommend that the maximum number be 4 digits
5. Also, it is needed to mention the structure and arrangement of MOF structures (use as reference: https://doi.org/10.1186/1752-153X-6-91) and the HOMO-LUMO gap, which is very important in electrochemical sensors (use as reference: https://doi.org/10.1021/jp405997r).
6. It was introduced in the last paragraph of the introduction that the paper refers to the adsorption mechanism of the gas sensing material for SF6. Please insert a picture of the proposed mechanism.
7. References: journal names need to be abbreviated. See https://www.mdpi.com/journal/nanomaterials/instructions#preparation
Minor errors: Page 14, figure 13, insert subscript in writing gas formulas.
Page 1 line 29, replace [1,2,3,4,5] with [1-5];
Page 1 line 32, replace [6,7,8] with [6-8]
Page 1 line 37, replace [11,12,13,14] with [11-14] and make it so along all the manuscript, where it is possible.
Based on these, I advise the authors to rectify the above-mentioned issues, and I hope to re-evaluate the revised manuscript.
Reviewer 2 Report
The manuscript titled "Research on the Adsorption Behavior of SF6 Decomposition Components by Mg-MOF-74" presents a comprehensive investigation of the adsorption characteristics of SF6, CF4, CS2, H2S, SO2, SO2F2, and SOF2 gases using Mg-MOF-74 through theoretical simulations. The findings are both thorough and compelling and deserve publication in Nanomaterials. The authors are advised to address the minor comment below before final acceptance:
- Authors should provide at least SF6 and CF4 adsorption studies on open metal MOFs, such as Cu-BTC or Zr-MOF (66 or 67), and inert metal site MOFs like Al or Zn-MOFs. Including SF6 and CF4 adsorption studies will strengthen the manuscript's conclusions for a broader audience.
- The SF6 adsorption studies will help differentiate between the chemical adsorption of CF4 and the physical adsorption of other gases, which will clarify the study's findings.
- Additionally, Both adsorption studies will provide insight into the role of the metal center and pore sizes in the MOFs.
Addressing these points will give the manuscript a stronger scientific foundation and appeal to a broader audience.
Reviewer 3 Report
The paper under review is devoted to the interesting issue of Mg-MOF-74 for gas sensing and identification of possible gas insulated switch gear faults. The authors have analyzed possible ways of some molecules' adsorption onto this MOF using quantum chemical methods. The calculations are performed at high level and enables the authors to get valuable results concerning interaction of some gases with MOF surface. The methodology of calculations seems to be absolutely adequate and the results obtained are reliable.
Nevertheless I would like to pay some attention to the issues questionable from common chemical knowledge.
1. First of all, I would like to recommend to modify the title of manuscript. I think that "Theoretical study" should be emphasized even in the title. Moreover, only SFn species are the products of decomposition of sulfur hexafluoride. All other species under discussion originate from the interaction of these rather unstable species with environment. So, "behaviour of SF6 decomposition components" do not reflect the real aspects of study.
2. I completely agree with the authors that the adsorption of the species resulting from SF6 decomposition products reactions with other compounds is not studied yet and this study is of practical interest. But why Mg-MOF-74 was chosen from the great variety of porous materials? I recommend the authors to motivate their choice.
3. The results of calculations depend on the size of cluster used. Therefore the dependence of calculation results from cluster size should be discussed.
4. It is of interest that different kinds of gas - surface interactions were established for different gases (physisorption for CF4 and chemisorption for other species). I would like to see some discussion about the reasons of this difference (actually, I expected to see physisorption for both CF4 and SF6 as very symmetrical molecules).
Reviewer 4 Report
The article is devoted to the quantum-chemical description of the different gases' interaction with Mg-MOF-74. These gases include SF6 molecules and the products of their decomposition after interaction with O2, H2O, and electrode materials, such as SF6, CF4, CS2, H2S, SO2, SO2F2, and SOF2 molecules. The authors considered the possibility to detect all of these gases with the help of Mg-MOF-74.
The article may be of some interest to other researchers, but it needs major revision in accordance with the statements below.
1. It is not clear from the introduction why the authors chose the Mg-MOF-74 structure in particular.
2. It is not clear why the authors optimized the geometry of the cluster extracted from Mg-MOF-74 in crystalline form. If the issue was to interpolate the results obtained for the cluster to the entire MOF structure, the extracted cluster should maximally correspond to a fragment of the entire structure. Its geometry relaxation, on the contrary, increases the difference in its structure from that of Mg-MOF-74, since the cluster is a molecule, and the entire MOF crystal has a periodic structure.
3. I believe that Figure 2 is redundant, since the structure of the gas molecules is known, and the structure of the Mg-MOF-74 cluster is shown in other figures.
4. The binding energy calculation according to Equation 1 is required to be carried out considering the basis set superposition error (BSSE), since this error, as is known, overestimates the energy values due to double counting of the basis functions.
5. The values of gas molecular energy, base energy before adsorption, and system energy after adsorption are redundant because they don't have much physical meaning. If a different DFT functional or basis set was used, these values would be different. Here only binding energy values make sense.
6. In Table 2, the authors present the van der Waals and covalent radii values. Is it the certain atoms' radii or the sum of the radii of the interacting atoms? An explanation is required here.
7. Tables 1-3 should be combined into one, which will collect all the parameters characterizing the interaction of gas molecules with Mg-MOF-74.
8. The manuscript contains a large fragment with a large number of tables (Tables 4-17) specifying the change in the geometry characteristics of gas molecules as a result of their interaction with Mg-MOF-74. These data are of no particular value. Therefore, they should either be deleted or moved to Supplementary materials.
9. In lines 377-378, the authors say that Figure 13 shows the HOMOs and LUMOs of Mg-MOF-74 before and after the adsorption of gas molecules. In this case, further discussions about how the localization of these orbitals changes are based on this figure. However, it follows from the caption to Figure 13 that it shows the orbitals after the adsorption of gas molecules. In this regard, it is difficult to verify the reasoning of the authors.
10. In lines 385-393, the authors give the values of the energy difference between HOMO and LUMO during the adsorption of different gas molecules, comparing them with Mg-MOF-74. In fact, this information duplicates what is presented in Figure 13. This fragment needs to be improved to avoid duplication.
The text requires intensive checking and correction of the English language. This is especially true for complex sentences in which punctuation marks are excessively used. These sentences should be divided into simpler ones. In addition, the text needs to be optimized. It needs to be presented more shortly and concisely. It presents too many details of the analysis, which do not affect the final result, but interfere with perception.
Round 2
Reviewer 1 Report
The authors have spent some effort to further improve the manuscript. They answered some of my questions well. Some minor points should be improved before it can be accepted for publication.
1. I observed that some the tables were moved to supplementary materials. It is good like this, but the numbering remained the same. You should compress Table 2 and Table 3 into one single table named: Table 2. Changes in bond length and bond angle before and after SF6 molecules adsorption Same procedure for Tables 4 and 5, and so on.
2. The text regarding the structure and arrangement of MOF structures as well as the HOMO-LUMO gap was improved, but the references recommended by the reviewer were not used. Therefore, read and cite the following manuscript: https://doi.org/10.1186/1752-153X-6-91, dealing with structure simulation into a supramolecular network and calculation of the metal ions/ligands ratio.
Please solve the proposed items, and then the paper can be published in Nanomaterials.
Reviewer 3 Report
The authors have modified the text according to my comments. I would like to pay attention only to two moments now:
1. I guess that some references supporting the choice of cluster for the calculations will be useful (that one periodic unit is enough to reproduce the real situation).
2. I think that it will be useful to add one-two sentences to show that this study really deals with NANOmaterials.
Reviewer 4 Report
The authors performed great work improving the text and answered all my questions addressed to them. However, there are some moments in the improved version which need a more precise definition.
1. Lines 98-99: The authors state that the accuracy of PBE0 hybrid functional is higher than on of LDA, GGA, and meta-GGA functionals. It needs to cite relevant works here because accuracy depends not only on functional type but mostly on the combination of functional and basis set as well as the system under study.
2. Lines 127-128: The authors should explain how EBSSE was calculated. It can be a direct explanation that includes operating by such term as “dummy atoms”, or at least it can be quoting the relevant works, where a method of EBSSE calculation is explained.
3. I recommend naming the tables in Supplemental material by adding any letter to the number, for example, “Table S1” instead of “Table 1”. The same numbering of the tables in the main text and Supplemental material makes confusion a little bit.
The first two sentences in the Abstract are too long. Each of them should be divided into a few ones. For example, the first one can be presented as:
“SF6 gas is an arc-extinguishing medium widely used in gas insulated switch gear (GIS). The insulation failure in GIS leads to the decomposition of SF6 in partial discharge (PD) and other environments. The detection of its main decomposition components is an effective method to diagnose the type and degree of discharge fault.”
Please, also check the whole text to improve such problems in the other places, if they are.
